# Short-Term Adverse Events and Antibody Response to the BNT162b2 SARS-CoV-2 Vaccine in 4156 Health Care Professionals

**DOI:** 10.3390/vaccines10030439

**Published:** 2022-03-13

**Authors:** Elena Azzolini, Lorenzo Maria Canziani, Antonio Voza, Antonio Desai, Jack Pepys, Maria De Santis, Angela Ceribelli, Chiara Pozzi, Massimo Turato, Salvatore Badalamenti, Luca Germagnoli, Alberto Mantovani, Maria Rescigno, Carlo Selmi

**Affiliations:** 1Department of Biomedical Sciences, Humanitas University, via Rita Levi Montalcini 4, Pieve Emanuele, 20072 Milan, Italy; elena.azzolini@humanitas.it (E.A.); antonio.voza@humanitas.it (A.V.); antonio.desai@humanitas.it (A.D.); yaakovyisrael.pepys@st.hunimed.eu (J.P.); maria.de_santis@hunimed.eu (M.D.S.); angela.ceribelli@hunimed.eu (A.C.); alberto.mantovani@hunimed.eu (A.M.); maria.rescigno@hunimed.eu (M.R.); 2IRCCS Humanitas Clinical Research Hospital, via Manzoni 56, Rozzano, 20089 Milan, Italy; lorenzo.canziani@humanitas.it (L.M.C.); chiara.pozzi@humanitasresearch.it (C.P.); massimo.turato@humanitas.it (M.T.); salvatore.badalamenti@humanitas.it (S.B.); luca.germagnoli@humanitas.it (L.G.); 3The William Harvey Research Institute, Queen Mary University of London, Charterhouse Square, London EC1M 6BQ, UK

**Keywords:** SARS-Cov-2, immunoglobulin, adverse event, real life

## Abstract

Short-term adverse events are common following the BNT162b2 vaccine for SARS-Cov-2 and have been possibly associated with IgG response. We aimed to determine the incidence of adverse reactions to the vaccine and the impact on IgG response. Our study included 4156 health-care professionals who received two doses of the BNT162b2 vaccine 21 days apart and obtained 6113 online questionnaires inquiring about adverse events. The serum response was tested in 2765 subjects 10 days after the second dose. Adverse events, most frequently a local reaction at the site of injection, were reported by 39% of subjects. Multivariate analysis showed that female sex (odds ratio—OR—1.95; 95% confidence interval—CI—1.74–2.19; *p* < 0.001), younger age (OR 0.98 per year, *p* < 0.001), second dose of vaccine (OR 1.36, *p* < 0.001), and previous COVID-19 infection (OR 1.41, *p* < 0.001) were independently associated with adverse events. IgG response was significantly higher in subjects with adverse events (1110 AU/mL—IQR 345-1630 vs. 386 AU/mL, IQR 261-1350, *p* < 0.0001), and the association was more pronounced in subjects experiencing myalgia, fever, and lymphadenopathy. We demonstrate that a more pronounced IgG response is associated with specific adverse events, and these are commonly reported by health care professionals after the BNT162b2 vaccine for SARS-Cov-2.

## 1. Introduction

Severe acute respiratory syndrome coronavirus 2 (SARS-CoV-2) infection and the resulting coronavirus disease 2019 (COVID-19) affected nearly 200 million people with over 5 million deaths as of December 2021, according to the World Health Organization (WHO). Effective vaccines are essential to limit the viral spread, and the Pfizer–BioNTech BNT162b2 vaccine based on a nucleoside-modified mRNA [1] was the first to be approved by both European and United States regulatory agencies [2] among the four currently approved vaccines [3,4].

While molecular proof of incident infections and the risk of developing severe forms of COVID-19 determine the efficacy of vaccines, serum levels of spike-specific IgG mirror the immune response [5]. Furthermore, short-term adverse events are common following each dose of the BNT162b2 vaccine, as evident from the reports on self-reporting apps [6] with a proposed impact on the IgG response [7].

We report herein the incidence of adverse events and the serological response 10 days after two doses of BNT162b2 vaccine in a cohort of 4156 health care professionals from a COVID-19 high-impacted hospital. We report that age, previous SARS-Cov-2 infection, and the occurrence of minor adverse events following vaccination are independently associated with a more pronounced IgG response to vaccination.

## 2. Materials and Methods

### 2.1. Subjects

We enrolled 4156 health-care professionals who received the two doses of the Pfizer-BioNTech BNT162b2 vaccine 21 days apart between 29 December 2020 and 1 April 2021. The study was coined *IgG-COVID* and was approved by the Humanitas Institutional Review Board, and all subjects signed informed consent prior to the blood draw. Following each dose of vaccine, all subjects were invited to fill out an online questionnaire reporting anthropometric data, including blood type [8], along with adverse events after the vaccine administration. Subjects were asked to indicate the presence of any symptoms and their durations using open questions; if any adverse event was present, data regarding drugs taken and emergency department admissions were asked. Additional comments were possible in every part of the questionnaire. In total, we included in the analysis 6113 questionnaires (3160 after the first dose, 2953 after the second dose). All reported events were included as possible vaccine-related adverse events.

### 2.2. Serum Tests

Five ml of peripheral blood was drawn ten days after the second dose from 2765 (67%) subjects. Samples were analyzed on the same day using the LIAISON^®^ SARS-CoV-2 S1/S2 IgG to determine the number of specific antibodies, IgG anti-S1 and anti-S2 [9].

### 2.3. Statistical Analysis

In the descriptive analysis, age, sex, available BMI, and blood group were reported per patient. Adverse events were reported per the individual questionnaire. The main differences between patients with or without reported adverse events were assessed with the Chi-squared test or Mann–Whitney test for categorical or continuous variables.

First, for all the reported adverse events, we performed a multivariable logistic regression model, using the presence of at least one reported adverse event as an outcome. Variables were selected if they showed a difference in the univariable model and were filtered if they had a high level of missing values (>10%). Second, for every subject, we performed a Mann–Whitney test to assess the interaction between different variables and the IgG titer. For continuous variables, we divided the population in two by median value. *p*-values lower than 0.05 were considered significant. Statistical analyses were performed with STATA version 16.0 (StataCorp LLC, College Station, TX, USA).

## 3. Results

### 3.1. Study Population

Our study population included 4156 health-care professionals from one tertiary care hospital in Northern Italy that was highly impacted during 2020 by the first wave of the COVID-19 pandemic. Of the 4156 subjects, 1589 (38%) were males, the median BMI was 23.3 (interquartile range—IQR—21.0–25.7), and the median age was 37 years (IQR 27–48). A total of 262 (44.6%) had blood type 0, 230 (39.1%) type A, 64 (10.9%) type B, and 32 (5.4%) type AB, while 491 (83.5%) were positive for the Rh factor. Previous COVID-19 was recorded in 843 (20%) subjects with a documented positive molecular test on the nasopharyngeal swab.

### 3.2. Adverse Events

The questionnaires reported a total of 2211 adverse events (1031 following the first dose and 1180 following the second dose), as shown in Table 1. 

The median number of symptoms reported by each subject was three (IQR 2–6), with generally fewer symptoms (median 2, IQR 1–4) reported after the first dose compared with the second dose (median 5, IQR 3–7). The overall mean duration of symptoms was three days. We observed that 971 (44%) participants reported having to take medications, 232 (23%) after the first dose and 739 (63%) after the second; in most cases, subjects reported taking antipyretics (98% overall, 94% and 99.8% following the first and second dose, respectively) whereas other drugs were used in only 38 cases (23 and 15 following the first and second dose, respectively). Eight subjects reported seeking medical attention at the Emergency Department (three following the first dose and five following the second dose), all of whom were discharged within a maximum of 4 h. Details on reported symptoms and their duration are summarized in Appendix A.

Female gender, younger age, lower BMI, a previous SARS-Cov-2 infection, and the second vaccine dose were significantly associated with reported adverse events (Table 2).

Only 614/2211 (28%) questionnaires with reported adverse events were compiled by males, compared with 1693/3916 (43.6%) questionnaires without adverse events (*p* < 0.001). The median age of participants with at least one adverse event was 35 (IQR 27–46) versus 38 (IQR 28–49) for those not reporting adverse events (*p* < 0.001). The BMI was lower in subjects reporting adverse events: 22.6 (IQR 20.4–25.0) vs. 23.4 (IQR 21.3–26.0) (*p* < 0.0001). Adverse events were reported more frequently following the second dose of the vaccine (53% vs. 45% after the first dose, *p* < 0.0001). Of the 843 questionnaires sent by subjects with a previous history of COVID-19, 375 reported an adverse event (45%, *p* < 0.001). 

A multivariable analysis using a logistic regression model for adverse events occurrence (Table 2) led to BMI being discarded because of missing values while female sex (odds ratio—OR—1.95; 95% confidence interval—CI–1.74–2.19; *p* < 0.001), age (OR 0.98 per year, *p* < 0.001), second dose of vaccine (OR 1.36, *p* < 0.001), and previous COVID-19 infection (OR 1.41, *p* < 0.001) reached statistical significance.

### 3.3. Antibody Response

The antibody response was measured 10 days after the second dose in 2765 subjects (990 males, 1775 females). The median IgG response was 856 AU/mL (IQR 294–1470), and data are illustrated in Table 3. 

Male subjects had a median of 432.5 AU/mL (IQR 238–1370), significantly lower than females (963 AU/mL IQR 302–1520, *p* = 0.0006). Participants 37-year-old or younger (n = 1452) had a significantly higher IgG response with a median of 1030 AU/mL (IQR 324–1540) compared to 465 AU/mL (IQR 268–1350) in subjects older than 37 (*p* < 0.0001). Subjects with a history of previous COVID-19 infection (n = 412) had a significantly higher IgG response with median titers of 2285 AU/mL (IQR 1195–3750) versus 421 AU/mL (IQR 286–1300) in subjects without previous COVID-19 (*p* < 0.0001). 

Of the 2765 subjects whose samples were analyzed, 1143 (41%) reported adverse events. This was associated with a median IgG response of 1110 AU/mL (IQR 345–1630), significantly higher when compared with subjects who did not report any symptoms following vaccination (386 AU/mL, IQR 261–1350). This difference was more relevant for muscle pain, fever, and lymphadenopathy (*p* < 0.0001).

## 4. Discussion

Our prospective cohort study on a large number of health-care professionals vaccinated with two doses of the BNT162b2 vaccine for SARS-Cov-2 demonstrates that self-reported adverse events are most frequently mild and correlate with sex, age, BMI, and previous COVID-19, and are more prevalent after the second dose, in partial agreement with previous reports [10,11]. Furthermore, the presence of adverse events, particularly fever, lymphadenopathy, and myalgia, predicts the magnitude of serum IgG response 10 days following the second dose.

We believe that the observations stemming from this study are important to the management of COVID-19 vaccination for several reasons. Our observations provide an estimate of overall self-reported adverse events in a large population of unselected health-care professionals who received two doses of the same vaccine with an identical schedule. The prevalence of adverse events following the first and second dose of vaccine was different compared with earlier reports, as represented by the variable prevalence of common systemic reactions such as fatigue, fever, and headache [1,12]. We cannot rule out the possibility that the self-reported source of data may be associated with the under-reporting by subjects without adverse events, but a similar technical approach has been used in other studies [6] and allows us also to identify events not requiring medical attention. On the other hand, self-reported data do not provide a solid estimate of the correlation between adverse events and the vaccine, for which the incidence of adverse events should be gathered from controlled trials [12], and we did not observe any serious adverse events within the first weeks following vaccination, in accordance with the major surveillance programs [13]. We are aware that the young age of this population may not adequately reflect the general population, particularly for older subjects in which the tolerability profile and the IgG response may differ significantly [14].

We performed serum tests for IgG titers 10 days after the second vaccine dose in the majority of the enrolled subjects, and our multivariate analysis demonstrated a correlation of the serological response with female sex, younger age, previous COVID-19, and specific adverse events. The previous COVID-19 has been associated with a higher IgG response to the vaccine [15,16], including differences in B memory cells [17], as well as the T cell response [18], and our data confirmed this observation with median IgG titer five times higher in subjects with previous infection, similar to previous reports [9]. 

Among the factors influencing the IgG response 10 days after the second vaccine dose, we observed that adverse events are significantly correlated with the antibody titer, as previously suggested by a smaller study on healthy subjects [19]. Other studies failed to identify a correlation between adverse events and the IgG titers one month following the BNT162b2 vaccination, but the same authors identified sex and body weight as predictors of the incidence of adverse events [20]. Our data confirm that female gender and younger age are significantly associated with adverse events, and this may reflect the more efficacious immune response observed in women for SARS [21] and other vaccines [22], particularly at a younger age. Among adverse events, lymphadenopathy has been observed in 78.4/100,000 subjects after the SARS-Cov-2 vaccination, more frequently axillary and unilateral [23,24]. This sign may require imaging when lasting for several weeks [25] or in patients undergoing oncological follow-up [26] and may lead, in some cases, to the diagnosis of hemophagocytic lymphohistiocytosis [27]. In our study, we observed a higher incidence of self-reported lymph node enlargement (4.4%) and a significant correlation with the IgG response. 

We are aware that our study has strengths and limitations. Among the strengths, we include a large number of enrolled subjects, the uniform vaccine type and schedule, and the established laboratory technique for IgG determination. The potential pitfalls of the study may derive from the self-reported origin of adverse events, which may lead to a less reliable representation of signs and symptoms and the young age of the participants, likely less representative of the general population [28]. In the case of self-reported items, however, the literature suggests that this approach may lead to an over-response by subjects who experienced adverse events, and this is likely to stress the safety profile of the vaccine [12,13,29]. Furthermore, we do not have details on the medical history of the subjects enrolled and cannot rule out the presence of ongoing immunosuppressants or coexisting comorbidities in the determination of adverse events or the serum response to vaccination.

## 5. Conclusions

In conclusion, we demonstrate that adverse events and the IgG response after the BNT162b2 vaccine for SARS-Cov-2 share major factors, including female sex, young age, and previous COVID-19 infection, thus supporting the immunological mechanisms of systemic side effects, particularly lymphadenopathy and fever. No safety issues emerged. We encourage further studies to determine the protection against COVID-19 in at-risk populations such as health care professionals.

## Figures and Tables

**Table 1 vaccines-10-00439-t001:** Self-reported adverse events following the two doses of vaccine. Continuous variables are expressed as median (IQR).

Events	Total	Dose 1	Dose 2
Number of events	2211	1031 (46.6%)	1180 (53.4%)
Number of symptoms	3 (2–6)	2 (1–4)	5 (3–7)
Duration of symptoms	3 (1–7)	3 (1–3)	3 (1–7)
≤1 day	733 (33.2%)	372 (36.1%)	361 (30.6%)
≤3 days	999 (45.2%)	476 (46.2%)	523 (44.3%)
≤7 days	350 (15.8%)	126 (12.2%)	224 (19.0%)
≤14 days	92 (4.1%)	37 (3.6%)	55 (4.7%)
≤21 days	37 (1.7%)	20 (1.9%)	17 (1.4%)
Emergency department access	8 (0.36%)	3 (0.29%)	5 (0.42%)
Any drug taken for adverse events	971 (43.9%)	232 (22.5%)	739 (62.6%)
- antipyretics	587 (98.2%)	165 (94.3%)	422 (99.8%)
- other drugs	38 (6.4%)	23 (13.1%)	15 (3.6%)

**Table 2 vaccines-10-00439-t002:** Factors associated with the occurrence of any adverse events following SARS-Cov-2 vaccination at univariate and multivariate analyses based on logistic regression.

	Univariable Analysis		Multivariate Analysis (n = 6111)	
	Odds Ratio (95% Confidence Interval)	*p* Value	Odds Ratio (95% Confidence Interval)	*p* Value
**Female gender**	1.98 (1.77–2.21)	<0.001	1.95 (1.74–2.19)	<0.001
**Age (years)**	0.98 (0.98–0.99)	<0.001	0.98 (0.98–0.99)	<0.001
**BMI**	0.95 (0.93–0.97)	<0.001		
**Blood group** **(vs. O)**				
**- A**	−0.99 (0.76–1.30)	−0.956		
**- B**	−0.80 (0.53–1.24)	−0.323		
**- AB**	−1.13 (0.64–2.02)	−0.662		
**Rh+**	0.79 (0.57–1.1)	0.167		
**2nd vaccine dose** **(vs. 1st)**	1.37 (1.24–1.52)	<0.001	1.36 (1.22–1.51)	<0.001
**Previous COVID-19**	1.51 (1.30–1.74)	<0.001	1.41 (1.22–1.65)	<0.001

**Table 3 vaccines-10-00439-t003:** IgG titers (AU/mL) following two doses of SARS-Cov-2 vaccine based on subject features and reported adverse events; the Mann–Whitney test was used.

		Median IgG (IQR)	*p* for Interaction
**Total**	(n = 2765)	856 (294–1470)	
**Gender**	Male (n = 990) Female (n = 1775)	432.5 (283–1370) 963 (302–1520)	0.0006
**Age (years)**	≤37 (n = 1452) >37 (n = 1313)	1030 (324–1540) 465 (268–1350)	<0.0001
**Blood groups**	O (n = 223) A (n = 188) B (n = 46) AB (n = 23)	956 (306–1430) 1080 (362–1635) 918 (306–1370) 1250 (371–1710)	
**Rh factor**	Positive (n = 402) Negative (n = 78)	1040 (347–1430) 996 (265–1630)	
**Previous COVID-19**	Yes (n = 412) No (n = 2353)	2285 (1195–3750) 421 (286–1300)	<0.0001
**BMI**	≤23.2 (n = 856) >23.2 (n = 811)	937 (312–1460) 719 (288–1460)	0.0540
**Any adverse event**	Yes (n = 1143) No (n = 1622)	1110 (345–1630) 386 (261–1350)	<0.0001
**Muscle Pain**	Yes (n = 640) No (n = 2125)	1200 (368–1825) 495 (278–1370)	<0.0001
**Fever**	Yes (n = 441) No (n = 2324)	1230 (374–1830) 637 (283–1390)	<0.0001
**Lymphadenopathy**	Yes (n = 197) No (n = 2568)	1300 (383–1880) 793 (289–1430)	<0.0001

## Data Availability

According to the policy of the hosting Institution, data will be made available through Zenodo.

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
