# Peer review of "Short-Term Adverse Events and Antibody Response to the BNT162b2 SARS-CoV-2 Vaccine in 4156 Health Care Professionals"

_vaccines, 2022, doi:10.3390/vaccines10030439_

Round 1

Reviewer 1 Report

This study reports data about the occurrence of side effects following the Pfizer-BioNTech mRNA vaccine against SARS-CoV-2, which show a link with age, gender and more importantly with the level of anti-Spike IgG produced. The paper is well written, the study seems adequately designed and the results presented are clear. The numbers and duration of symptoms show substantial inter-individual variability. Was there any association of the studied factors and of IgG levels with these parameters rather that just with occurrence of any adverse event?

Reviewer 2 Report

In this report in >4000 health professionals the IgG response is measured 10 days after administration of the 2nd doses of BNT162b2 vaccine. Furthermore minor adverse events are registered and they were shown to be independently related to a more pronounced IgG response.

Methods described are relatively simple. However it is unclear when which test is applied. 

Results: table 1: data are clear, however some clarification is needed, ie “any drug” presents the number of patients that took a medication. The table should be readable without the text of the paper itself.

Table 2: why are these factors chosen? Please clarify.

Which statistical test is used for the analyzing the data? In the methods the tests are summarized, however it is not clear which test is used in a specific case.

In table 3 IgG titers are related to patient characteristics. It is not clear why these characteristics are chosen. Please clarify.

Since many analyses are performed in the same dataset, results should be corrected for multiple testing.

Discussion: does the height of the IgG titers is in accordance with previous studies?

Reviewer 3 Report

Major comments: When a short-term adverse event is recorded, it may or may not be caused by vaccination. In this study, there is not only no control group, but no baseline data, comparator population, etc. Because of this study limitation, the study does not provide an estimate of "specific" adverse events (l. 159), and the statement that the safety profile is confirmed should be appropriately qualified by the way in which the safety profile is confirmed (e.g., putting a bound on the adverse events rather than demonstrating adverse events are not elevated from baseline in a major way), and the use of "Determinants" in the title would be better as "Correlates" because the causal nexus has not been elucidated.   ll. 189-192: Failure to achieve statistical significance is not sufficiently strong evidence of absence of an effect. An equivalence test was not performed. This claim is unsupported.   Minor: l 36 casualties: ambiguous in English because it can mean morbidity or mortality. rephrase l 42 reference [6] does not justify "immune response and protection" claims due to its study design. It is unclear what is meant by "an assumption for which definitive evidence is awaited."  
